# Limits of Detection of Mycotoxins by Laminar Flow Strips: A Review

Xinyi Zhao [1,2], Hugh J. Byrne [2], Christine M. O'Connor [1], James Curtin [3] and Furong Tian [1,2,*]

1   School of Food Science Environmental Health, Technological University Dublin, D07 ADY7 Dublin, Ireland; d20127084@mytudublin.ie (X.Z.); christine.oconnor@tudublin.ie (C.M.O.)
2   FOCAS Research Institute, Technological University Dublin, D08 CKP1 Dublin, Ireland; hugh.byrne@tudublin.ie
3   College of Engineering and Built Environment, Technological University Dublin, D01 K822 Dublin, Ireland; james.curtin@tudublin.ie
*   Correspondence: furong.tian@tudublin.ie

**Abstract:** Mycotoxins are secondary metabolic products of fungi. They are poisonous, carcinogenic, and mutagenic in nature and pose a serious health threat to both humans and animals, causing severe illnesses and even death. Rapid, simple and low-cost methods of detection of mycotoxins are of immense importance and in great demand in the food and beverage industry, as well as in agriculture and environmental monitoring, and, for this purpose, lateral flow immunochromatographic strips (ICSTs) have been widely used in food safety and environmental monitoring. The literature to date describing the development of ICSTs for the detection of different types of mycotoxins using different nanomaterials, nanoparticle size, and replicates was reviewed in an attempt to identify the most important determinants of the limit of detection (LOD). It is found that the particle size and type of materials contribute significantly to determining the LOD. The nanoparticle sizes used in most studies have been in the range 15–45 nm and gold nanoparticle-based ICSTs have been shown to exhibit the lowest LOD. Perspectives for potential future development to reduce the LODs of ICSTs are also discussed.

**Keywords:** nanoparticle; particle size; mycotoxins; aflatoxin; fusarium toxins



## 1. Introduction

Mycotoxins are toxic natural secondary metabolites produced by fungi, for example *Aspergillus* and *Fusarium*, commonly found on agricultural commodities in the field or during storage. These toxins cause food and feed-borne intoxication, and many are cytotoxic, carcinogenic, mutagenic, or immunosuppressive [1,2]. Aflatoxin is produced by *Aspergillus (A.) flavus* and *A. parasiticus*, and exists as three major types: B1, B2, G1, and its hazardous effects have been well documented [1]. Fusarium toxins are produced by Fusarium, and are classified in three major groups: Zearalenone (ZEN), Fumonisins, and Trichothecenes [2]. ZEN is a non-steroidal estrogenic compound with a toxic estrogen effect, destroying the reproductive system of animals, resulting, for example, in estrogen syndrome in pigs, despite its low toxicity after oral administration [3]. Fumonisins have a strong structural similarity to sphinganine and elicit toxic estrogen effects which can lead to cancer [4]. Fumonisins can be divided into four main subgroups, A, B, C, and P, and a subtype of fumonisin B, fumonisin B1 (FB1), is the most toxic and abundant of all the fumonisins [5]. Trichothecenes are a class of sesquiterpenes and also have toxic estrogen effects on the reproductive performance of animals and humans. Trichothecenes have four main subgroups: A, B, C, and D, and the subtype of Trichothecenes A (T–2) is the most common, presenting a potential hazard to human health worldwide [6].

Mycotoxins usually appear in crops and food produce such as maize, sorghum, wheat, oats, rice, soybean, sunflower, cotton seeds, chili peppers, black pepper, coriander, turmeric,

ginger, peanut, pistachio, almond, walnut, coconut, Brazil nut, dry vine fruits, wine and grape juice, and rice liquor, mainly due to the high moisture content and temperatures suitable for the growth of fungi [7]. Field drying has been an accepted practice since commercial farming began, but it depends on sun and wind, and mechanical drying is often needed, to reduce the moisture content to 12–14% in a crop to slow the growth of fungi [8]. The moisture content and the concentration of mycotoxins of the grains are measured continuously, from when the grains are harvested to when they are sold to the consumers [9]. Most common quantitative methods are hand oven tests and moisture meter tests [10]. Due to the health risks for humans and animals, authorities such as the European Commission or the US Grain Inspection, Packers and Stockyards Administration (GIPSA) have attempted to address the mycotoxin problem by adopting regulatory limits. Regulations are in force for, e.g., aflatoxins and fusarium toxins in selected foodstuffs (EC 1881/2006) [11], and there are recommendations for maximum levels of mycotoxins in feed (EC (2006/576/EC)) [12]. Among aflatoxins, Aflatoxin B1 (AFB1) is the most toxic as a potent carcinogen and many countries have implemented maximum residue limits (MRLs) of AFB1 [13]. In China the "GB 2761-2017 Food Mycotoxin Limit" strictly sets the MRL of AFB1 in wheat, wheat flour, corn, and corn flour to 20 ng/mL [13]. Among all fusarium toxins the ZENs are the most strongly associated with chronic and fatal toxic effects in animals and humans [5,6]. In Italy the MRL of ZEN in cereals and cereal products is 100 ng/mL [14], and 50 ng/mL in Australia [15].

Currently, rapid portable testing platforms for the detection and quantification of potentially dangerous mycotoxins in food and beverage production are very limited. The most used techniques for detecting AFLs are thin layer chromatography (TLC) and high-performance liquid chromatography (HPLC). However, these methods require extensive sample preparation, expensive instruments, and professional operation. Alternatively, enzyme-linked immunosorbent assay (e.g., ELISA) has been successfully developed for AFLs [16]. However, ELISA also requires incubation and washing steps which are mainly confined to laboratories [17]. Thus, there is a need for rapid, cost-effective, accurate testing techniques for definitive fingerprint detection of disease and environmental markers at low concentrations.

Technologies such as microplate readers are based on 'non-traceable' approximated methodologies and rely on time-consuming sample preparation and cleaning [18,19]. Existing testing techniques, such as immune-based and molecular assays, e.g., ELISA and polymerase chain reaction approaches, provide accurate and sensitive detection of mycotoxins [19–21]. However, these methods are labor and time-intensive (normally requiring tens of people and several hours) [20–23].

Alternative and more rapid methods are not accurate and often give false positive/negative results [24]. Thus, there is a need for rapid, cost-effective, accurate testing techniques for detection of mycotoxins to ensure the safety and quality of food. Microfluidics, Lab-on-a-Chip, Smart Nanospectroscopy, Lab-in-a-Fiber, sensor technologies are among the key technological interfaces that have been optimized from micro- to nano-scales [24–26]. Microfluidic/Optofluidic lab-on-a-chip technologies are a commonly used solution in the detection of mycotoxins, particularly point-of-care testing; integrating functional modules commonly used in laboratories onto a small chip, for biomarker testing [26,27]. The technology provides a unique characteristic of high detection surface area to volume ratio for a short analysis time, enabling complex diagnostic assays [27,28]. However, 'interconnect and read-out bottleneck' and heat removal are two widely acknowledged challenges limiting performance [29].

Lateral flow immunochromatographic strips (ICSTs) have received increasing attention for qualitative and quantitative analysis in different areas [17]. For example, deoxynivalenol strips from Neogen Corp have been widely used in food safety and environmental monitoring [30]. Sandwich structures are usually used in large-scale detection, while sensitive giant magnetoresistive-based immunoassays are used for multiplex mycotoxin detection [31]. Different nanomaterials have been conjugated with antibodies to enhance detection limits

for mycotoxins. Delmulle et al. (2005) developed an ICSTs with 40 nm gold nanoparticles for the detection of Aflatoxin B1 in pig feed [7]. Liao and Li (2010) subsequently devoted significant effort to investigating the effect of core-shell silver-gold nanocomposites on the properties of ICSTs [32]. However, these detection methods can only provide qualitative results (positive or negative) or semi-quantitative information on analyte concentration, and the ICSTs could not satisfy the requirements for practical applications [32]. Therefore, many ICSTs devices providing quantitative analyte concentration testing have been subsequently developed which have improved the sensitivity and specificity for mycotoxin detection and quantification (normally 1–20 ng/mL) [33]. However, due to the bulkiness of their readout systems, their applications are limited [34,35]. Obviously, an ICST reader which is based on a mobile device would be advantageous, as it would satisfy the requirements of portable and feature-rich testing. The mobile health market is rapidly developing and portable diagnostics tools provide an opportunity to increase the availability of healthcare and decrease costs [36]. This paper reviews the developments of ICSTs' technology for different types of mycotoxins, the influence of different materials for nanoparticles, different sizes of nanoparticle, and replication on the limit of detection (LOD) for mycotoxins, and their adaptation to mobile and/or portable readout systems.

## 2. Research Methods

Information was collected in Google Scholar and Science Direct with keywords: mycotoxin, strip test, LOD. Twenty-five peer-reviewed papers from 2006–2021 were compared to identify the lowest LOD for different mycotoxins (Aflatoxin B1: 10; Zearalenone: 5; Fumonisin B1: 5; Trichothecene–A: 5) in Figure 1. Please note that some papers included the LOD of more than one mycotoxin, but only the mycotoxin with the lowest LOD in that paper is discussed here. The data were collated by type of mycotoxin, type of materials for nanoparticle, particle size (nm), number of replicates, and year (Table 1).

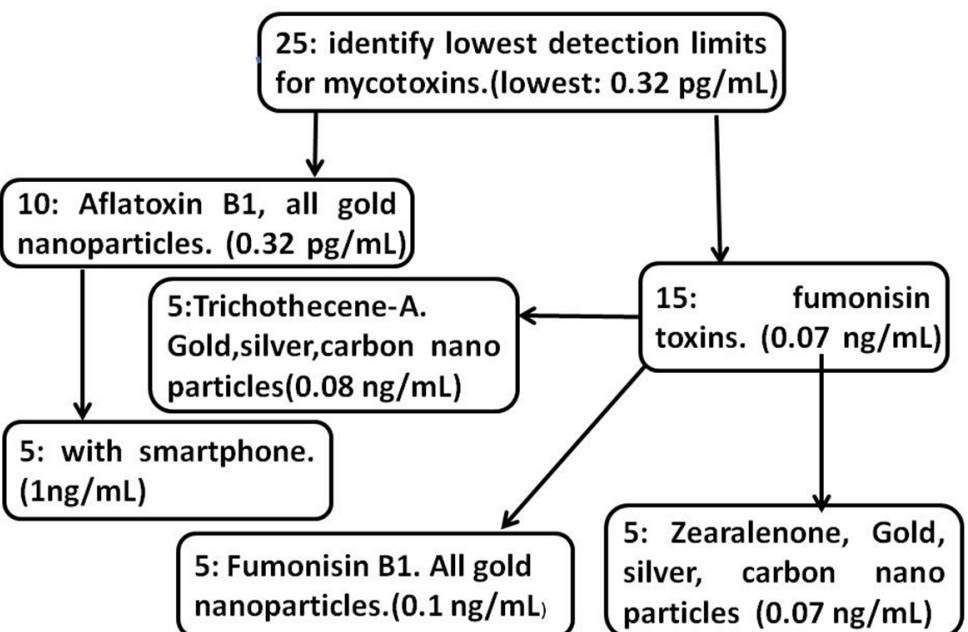

**Figure 1.** Hierarchy of analysis of papers dealing with limits of different mycotoxins.

Statistical analysis: ANOVA with a non-parametric technique (Kruskal–Wallis rank sum test), Dunn's test (post-hoc) and multilinear regression were employed to analyse the significant determinant parameters of LOD.

**Table 1.** Papers with limit of detection of mycotoxins.

| Type | Particle | Size (nm) | Year | Replicates | Standard Deviations | Coefficient of Variation (%) | LOD (ng/mL) | Reference |
|------|----------|-----------|------|------------|---------------------|------------------------------|-------------|-----------|
| AFB1 | Gold | na | 2013 | na | na | na | 0.1 | [10] |
|  | Gold + phone | 14 | 2019 | 6 | 0.01 | 2.7 | 0.3 | [36] |
|  | Gold | 45 | 2015 | 6 | 0.09 | 9.47 | 0.00032 | [37] |
|  | Gold | na | 2014 | na | na | na | 0.03 | [38] |
|  | Gold + phone | 30 | 2019 | 6 | na | 1.5 | 2 | [39] |
|  | Gold | 25 | 2006 | 11 | 0.44 | 10.4 | 2.5 | [40] |
|  | Gold | na | 2013 | 3 | na | na | 2 | [41] |
|  | Gold + phone | na | 2019 | 9 | na | na | 2 | [42] |
|  | Gold + phone | 1.4 | 2017 | 5 | na | na | 3 | [43] |
|  | Gold + phone | 40 | 2013 | 5 | na | na | 5 | [44] |
| FB1 | Gold | na | 2007 | 12 | 0.3 | 6.67 | 0.1 | [45] |
|  | Gold | na | 2014 | 5 | 0.18 | 6.01 | 0.12 | [46] |
|  | Gold | 30 | 2017 | 3 | na | na | 0.6 | [47] |
|  | Gold | 20 | 2020 | na | na | na | 2.5 | [48] |
|  | Gold | 40 | 2014 | 3 | 0.9 | 10.6 | 5 | [49] |
| ZEN | Gold | 25 | 2017 | 3 | 0.16 | 6.2 | 0.07 | [50] |
|  | Gold | 17 | 2020 | 5 | 0.09 | 4.9 | 0.25 | [51] |
|  | Silver | na | 2018 | 3 | na | na | 0.25 | [52] |
|  | Silver | 15 | 2018 | 10 | 0.22 | 4.85 | 0.58 | [53] |
|  | Carbon | 190 | 2016 | 3 | 0.05 | 3.79 | 12 | [54] |
| T–2 | Gold | na | 2019 | 2 | na | na | 0.08 | [55] |
|  | Gold | 30 | 2017 | 3 | na | na | 0.1 | [56] |
|  | Silver | na | 2015 | na | na | na | 0.9 | [57] |
|  | Silver | 45 | 2017 | na | na | na | 5 | [58] |
|  | Carbon | 120 | 2017 | na | na | na | 13 | [59] |

Note: na means not applicable/unknown.

## 3. Results

Table 1 provides a breakdown of the analysis of the identified 25 peer-reviewed papers, according to the species of mycotoxin, the nanoparticle used, and the nanoparticle size, replicates and, where available, the reported LOD.

The ranges of the limits of detection are shown in Figure 2, according to (a) different mycotoxins and (b) different nanoparticle-based ICSTs, in an attempt to elucidate which one of them has a low limit of detection generally. Figure 2a indicates that 40% of articles focused on Aflatoxin B1, which is the most studied mycotoxin. A further 20% of the articles focused on Fumonisin B1, 20% of the articles focused on Zearalenone and 20% of the articles focused on Trichothecene–A. Figure 2b indicates that gold nanoparticles are the most frequently studied nanoparticle in the reviewed literature.

Figure 3 presents the timeline of the LOD for (a) four mycotoxins and (b) type of nanoparticle, across the 25 papers from 2006 to 2021. The graphs in Figure 3 show a positive trend with time, with a peak in 2017 for both parameters. The results indicate that, although there has been a growing interest in mycotoxin detection over the years from 2006 to 2021, the LOD has not decreased significantly over the time period. Gold-based nanoparticles have been the most popular from the outset. Notably there has been increased interest in alternative materials in more recent years, although they have not produced a significant reduction in the LOD.

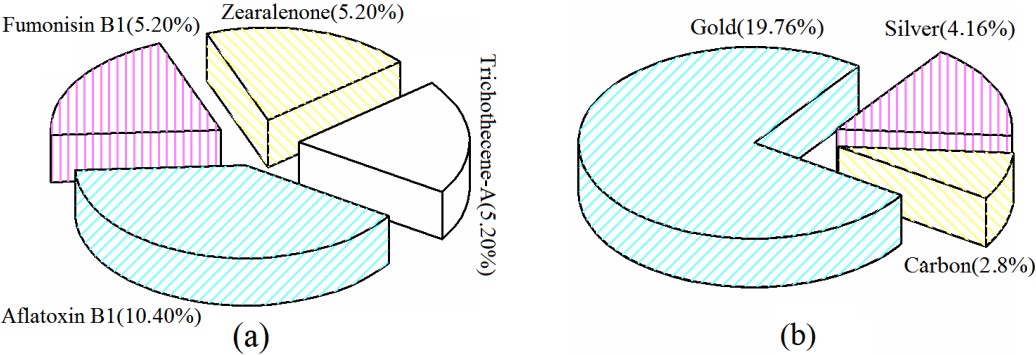

**Figure 2.** Percentage of studies which (**a**) addressed the different types of mycotoxins (Blue: Aflatoxin B1; Purple: Fumonisin B1; Yellow: Zearalenone; White: Trichothecene–A); and (**b**) the nanomaterials used in the ICST design (Blue: Gold; Purple: Silver; Yellow: Carbon).

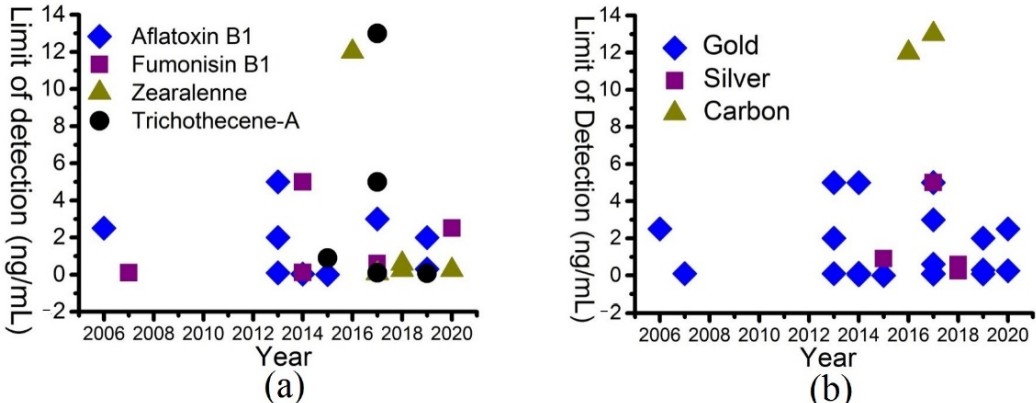

**Figure 3.** Timeline of limit of detection on mycotoxins and materials of nanoparticles. (**a**) Different mycotoxins (Blue diamond: Aflatoxin B1; Purple square: Fumonisin B1; Green triangle: Zearalenone; Black circle: Trichothecene–A); (**b**) Different materials of nanoparticles (Blue diamond: Gold; Purple square: Silver; Green triangle: Carbon).

Figure 4a presents the dependence of the LOD on nanoparticle size across the 19 papers for four mycotoxins. Figure 4b presents the dependence of the LOD on sizes of the different type of nanoparticle across the 19 papers. It can be seen that the data are predominantly clustered within a range of particle size from 15 nm to 45 nm, and the LOD from 0.1 to 5 ng/mL. Although Figure 4b indicates two outlying measurements for Zearalenone and Trichothecene–A, other measurements of these fall within the cluster, and Figure 4b indicates that these outliers are due to the relatively poorer performance of carbon nanoparticle-based devices.

Figure 5 presents the average LOD of different types of mycotoxins. Normally, the LOD is the lowest for gold nanoparticles, and it is followed by silver and the highest average LOD of strips is for carbon-based ICSTs. EU limits for Aflatoxin B1, Fumonisin B1, Zearalenone, and Trichothecene–A are 5, 500, 100, 25 ng/mL, respectively [11]. All of the LODs in this review paper are far lower than the EU limits.

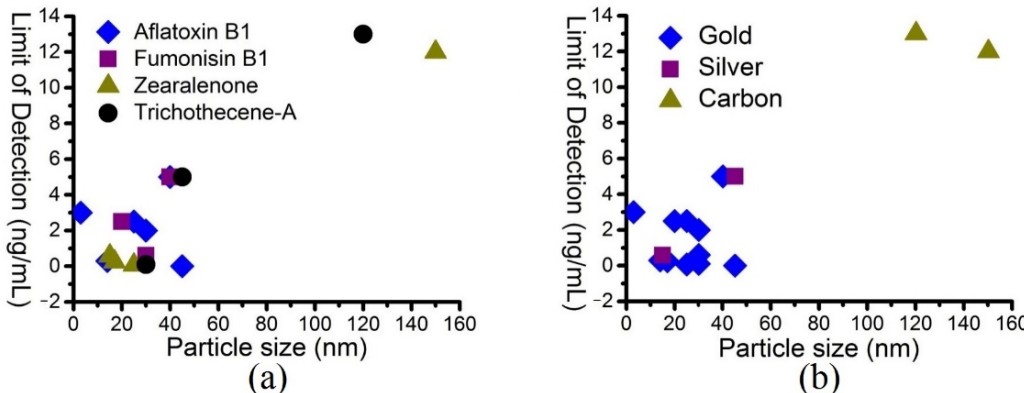

**Figure 4.** Nanoparticle size dependence of limit of detection on mycotoxins and materials of nanoparticles. (**a**) Different mycotoxins (Blue diamond: Aflatoxin B1; Purple square: Fumonisin B1; Green triangle: Zearalenone; Black circle: Trichothecene–A); (**b**) Different materials of nanoparticles (Blue diamond: Gold; Purple square: Silver; Green triangle: Carbon.).

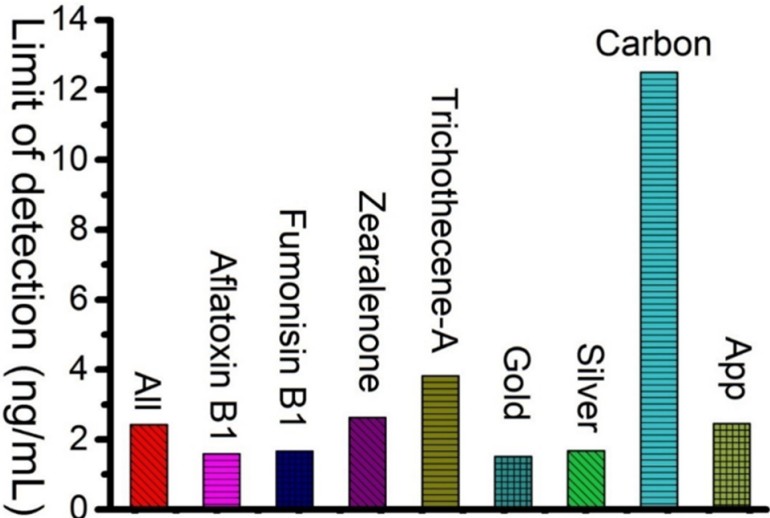

**Figure 5.** Average limits of detection of strips of different types of mycotoxins.

Since there are missing values in the datasets of particle size and replications, Mycotoxin, Nanoparticle, and Phone were analyzed by three-way ANOVA. From these three variables, only one variable (Nanoparticle) significantly contributes to the outcome ($p = 1.75 \times 10^{-7}$). The ANOVA results with a non-parametric technique (Kruskal–Wallis rank sum test) and Dunn's test (post-hoc) indicate that the significant difference is between the Nanoparticle groups, gold and silver. The ANOVA results with a non-parametric technique (Kruskal–Wallis rank sum test) and Dunn's test (post-hoc) indicate that the significant difference is between the gold Nanoparticle groups and the no gold Nanoparticle groups. There is no significant statistical difference between the LOD as read by a mobile phone or otherwise ($p = 0.153$).

Multivariate linear regression was performed to explore the relationship between the LOD and the parameters of type of mycotoxin, type of nanoparticle, particle size, year of publication, and replication. The linear regression had an $R^2 = 0.866$, particle size on prediction of the LOD ($R^2 = 0.904$), and particle type on prediction of the LOD ($R^2 = 0.901$). The resultant prediction equation is: LOD = 783.5 − 1.218 × Mycotoxin + 5.018 × Nanoparticle type + 0.0007274 × Particle size − 0.3875 × Year − 0.6992 × Replicates. Nanoparticle size is the strongest predictor ($p < 0.0001$, $R^2 = 0.904$), followed by Particle type and Replicates ($p < 0.05$).

## 4. Discussion

Figure 2a illustrates which Aflatoxin B1s were mentioned the most. These were Fumonisin B1, Zearalenone, and Trichothecene–A, in descending order. Gold nanoparticles are the most studied material, compared to silver and carbon (Figure 2).

The results indicate that there has been a growing interest in mycotoxin detection by ICST over the years from 2006 to 2021 (Figure 3). This growing interest is possibly due to nanotechnology development in monitoring the quality of the stored foods [60]. The majority of the papers with low LOD have involved gold nanoparticles (Table 1). In addition, the average LOD of gold is much less than those for silver and carbon nanoparticle-based devices. It can be easily seen from Figure 4a,b that the sizes of most nanoparticles are within 15–45 nm and their LODs are relatively low. The most popular nanoparticle size range for detection mycotoxins are gold nanoparticles in the range from 15 to 45 nm (Figure 4). The statistical analysis shows that the particle size is the highest determinant of the LOD ($R^2$ = 0.904), the second being particle type. The size of gold nanoparticle and the antibody-to-gold nanoparticle ratio can be precisely controlled. At the same time, their low toxicity has been confirmed in human cells and fungi [61–63]. In addition, silver ions have been found to be toxic to zebrafish, bacteria, and mouse stem cells [62–67]. Therefore, gold nanonanoparticles are a better choice for immunochromatographic test strip applications.

It can also be seen that the LOD of Aflatoxin B1 is much lower than the values attained for Fumonisin B1, Zearalenone, and Trichothecene–A. However, the average values for Zearalenone and Trichothecene–A are significantly influenced by the carbon nanoparticle-based measurements (Figure 4a). The type of mycotoxin is not a strong determinant of the LOD ($p$ = 0.118). The recommended maximum levels of Aflatoxin B1, Fumonisin B1, Zearalenone, and Trichothecene–A in feed by EC Regulations are 5, 500, 100 and 25 ng/mL, respectively, and the reported values for the LOD of the ICSTs fall within these limits.

This review also shows that the LOD with smartphone-based readout systems is similar to that by other means ($p$ = 0.153), although the data available to date are sparse. The smartphone app is a promising method used in the detection of quantitative concentration of mycotoxins by lateral flow immunochromatographic strip, and increasing development will provide enough data in literature to compare the detection efficiency.

The data in this paper are from 2006–2021, but the majority are from 2013–2021. Figure 3a,b show that the LOD for mycotoxin has not decreased continuously over the last few years. The year is therefore not a determinant of the LOD ($p$ = 0.153).

In addition, lowering the LOD may not be the only purpose because the LOD in this review paper are all within the requirements of the EU. It is important to fully characterize the analytical performance of reliable mycotoxin detection in order to understand its capability and limitations, and to ensure that it is "fit for purpose". For example, the LOD may well reside at some concentration below the linear range of an assay. A traditional and typical approach to estimate the LOD consists of measuring replicates, usually $n$ = 20, of a zero calibrator or blank sample, determining the mean value and SD, and calculating the LOD as the mean +2 SD [68]. To establish these parameters a manufacturer would use gold nanoparticles of a size within the range from 15 nm to 45 nm and test 20 sample replicates to increase the robustness and the statistical confidence of the estimate.

The complex sample matrix can strongly suppress the ICST response signal, to the detriment of the analytical performance [69]. To address such limitations and challenges, sample enrichment and assay improvement, mycotoxins can be bioconjugated with multimodal nanostructures for sample processing [70–72]. The analytes in a sample can be preconcentrated and/or amplified to enhance the detection limit [69]. Different mycotoxin single strips have been commercialised. It is often observed that different mycotoxins coexist in a single sample. The multiplexed detection adds a new dimension to increase the efficiency of mycotoxin testing [73]. Active materials such as magnetic particles, quantum dots, fluorophores, and nanoparticles can be conjugated with antibodies to enhance the ICST responses [64].

A schematic overview of the strategy for measuring various types of mycotoxins using multiplex enrichment for samples, various nanoparticles based on laminar flow strips and using a smartphone readout is shown in Figure 6. It suggests strategies such as: (i) the use of multimodal nanoparticles conjugated with mycotoxin antibodies to enrich the concentration of mycotoxin in the samples; (ii) the use of various labels (e.g., magnetic particles, quantum dots, fluorophores on gold nanoparticles) which may be viable options for multiplexed ICSTs; (iii) development of smartphone applications and data transfer, whereby the colours of the test lines are analyzed and converted into concentrations to provide advanced decision making.

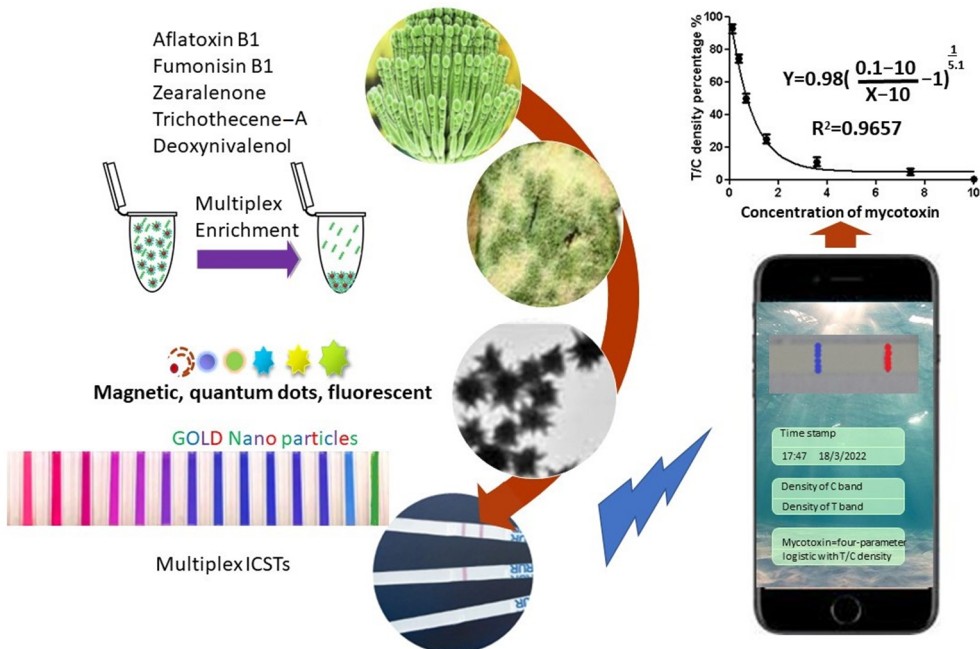

**Figure 6.** Schematic diagram of measurement protocol for various types of mycotoxins using multiplex enrichment for samples and various nanoparticles, based on laminar flow strips and using a smartphone.

## 5. Conclusions

The development of nanotechnology for monitoring the quality of stored foods has provided promising tools for improved quantitative performance. In order to improve the accuracy and precision of ICST, different parameters such as type of mycotoxin, type of materials, nanoparticle size, replicates, and use of smartphone readouts have been considered as determinants of the limit of detection (LOD). The results show that the type of nanomaterial, particle size, and number of replicates strongly contribute to predict the LOD. Immunochromatographic test strips based on gold nanoparticles within the range 15–45 nm have the lowest LOD. This review provides guidance for future developments of mycotoxin monitoring technologies, based on the enrichment of mycotoxins from samples, choice of magnetic, quantum dots, or fluorophores on gold nanoparticles for multiplexing ICST, and the development of machine learning for smartphone Apps.

**Author Contributions:** Conceptualization, X.Z. and F.T.; methodology, X.Z.; software, F.T.; validation, X.Z.; formal analysis, F.T.; investigation, X.Z.; resources, F.T.; data curation, F.T.; writing—original draft preparation, X.Z.; writing—review and editing, H.J.B., F.T. and C.M.O.; visualization, X.Z.; supervision, F.T., C.M.O. and J.C.; project administration, F.T.; funding acquisition, F.T. All authors have read and agreed to the published version of the manuscript.

**Funding:** This research was funded by the Researcher Award from TU Dublin postgraduate school.

**Institutional Review Board Statement:** Not applicable.

**Informed Consent Statement:** Not applicable.

**Data Availability Statement:** This study did not report any data.

**Conflicts of Interest:** The authors declare no conflict of interest.

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
