# Peer review of "Limits of Detection of Mycotoxins by Laminar Flow Strips: A Review"

_2673-3501, doi:10.3390/applnano3020006_

Round 1

Reviewer 1 Report

The manuscript entitled Limits of detection of mycotoxins by laminar flow strips: A Review submitted by the group of authors represents an extensive review on the topic of LODs of the lateral flow immunochromatographic strips (ICSTs) on detection of mycotoxins. Overall, the LODs were not significantly decreased in recent years, and only the use of nanoparticles has resulted in lower LODs.

The introduction part is fluent and follows the main objective.

The Authors should mention Reveal® Q+ MAX for DON (PN 8388, Neogen Corp.) deoxynivalenol strip.

The Authors could include images of the available strips and designs to give a reader the full perspective on the working principles of each approach.

In general, the manuscript is well elaborated and concentrate on the LODs of the mycotoxins strips.

The comment section gives a brief information on the topic and includes a critical points for future perspectives development of strips for mycotoxin.

Overall, the manuscript should be enriched with proposed comments and information. The manuscript is and average presentation on the topic that could in future have more importance if new ways of how to reduce the LODs would be found.

Author Response

Dear reviewer:

We thank the expert referee for valuable feedback, insightful comments and for giving the time to review our work. Following your suggestions, we have included significant additional information and clarified key points. Our point-by-point response below addresses their specific comments. The response to review 1 are red. All the changes are in the manuscript and locations are indicated in the responses.

Furong

Reviewer 2 Report

1. What does x-axis mean in Figure 2a,b?

2. For readers who are new to it, I think you need a scheme that encompasses the title and content. In other words, it would be nice to have a schematic diagram of measuring various types of mycotoxin using various nanoparticles based on laminar flow strips and using a smartphone.

3. I hope that you can suggest the future direction for the development of a biosensor for detecting mycotoxin with the highest performance. For example, it is possible not only to optimize particle types and sizes, but also to propose multiplexed detection, and to suggest strategies considering machine learning to further improve LOD.

Author Response

Dear Reviewer:

We thank the expert referee for valuable feedback, insightful comments and for giving the time to review our work. Following your suggestions, we have included significant additional information and clarified key points. Our point-by-point response below addresses their specific comments. The response to review 1 are blue. All the changes are in the manuscript and locations are indicated in the responses.

Best regards

Furong

Round 2

Reviewer 2 Report

Well addressed all the points suggested by the reviewer. I recommend accepting this manuscript.